# Virtual Educational Intervention of Craftswomen Working with Native Peruvian Cotton during COVID-19 for Reactivating the Artisian Tourism

**Rosse Marie Esparza-Huamanchumo** [1,*] , **Rosa Jeuna Diaz-Manchay** [2] **and Maribel Albertina Díaz-Vásquez** [2]

1 Facultad de Administración Hotelera, Turismo y Gastronomía, Universidad San Ignacio de Loyola, Lima 15024, Peru
2 Faculty of Medicine, Universidad Católica Santo Toribio de Mogrovejo, Chiclayo 14012, Peru
* Correspondence: resparza@usil.edu.pe; Tel.: +51-979111143

**Abstract:** The pandemic has significantly affected the tourism sector worldwide; however, craftswomen are a vulnerable group that has been affected economically by this crisis. This research evaluated the level of compliance with preventive measures before and after carrying out the virtual educational intervention for craftswomen working with native cotton in the Lambayeque Region, Peru. The methodology applied was a pilot study, quasi-experimental, without a control group. The population consisted of 30 craftswomen from the populated areas of La Raya–Túcume, Pómac III-Pitipo and Jotoro-Jayanca. SPSS Statistics v25 was used for data processing. The χ2 test was used in order to evaluate the variation before and after the intervention. The correlational findings demonstrate that after applying the virtual educational intervention, it is sufficient to apply specific measures in the first (before) and second stage (during) to obtain a higher result in compliance with the general level of the regulations against COVID-19. It is concluded that the virtual educational intervention for the craftswomen has generated awareness, impacting the care of their personal health, their family and their community, as well as being prepared for the reactivation of tourism.

**Keywords:** craftswomen; COVID-19; virtual educational intervention; tourism reactivation; sustainable development

## 1. Introduction

After nearly 25 years of declining global poverty, the outbreak of COVID-19 represents a grim setback for human development, further undermining the goal of reducing global extreme poverty to less than 3% by 2030 [1]. The tourism industry reduces poverty and inequality and has become one of the strategic and potential sectors in economic development at the regional, national and global levels [2]. Peru was one of the most affected South American countries in the first two pandemic waves of COVID-19, which hindered the reactivation of its tourism sector due to the sanitary measures imposed by the government, such as quarantine, physical distancing, etc. [3]. As preventive measures to prevent COVID-19, the World Health Organization [4] recommended the following: keeping 1 m away from others, using masks, frequent hand washing, covering the mouth and nose with the forearm or a disposable handkerchief when sneezing or coughing, and becoming vaccinated. For the tourism sector, this pandemic affects the most vulnerable groups, whose means of subsistence are the use of landscape resources or the sale of their products/services to visitors [5]. After agriculture, the cottage industry is the mainstay of the rural economy. However, it is characterized by having a separate market, as both artisans and customers are dispersed [6]. Specifically, the making of textiles with native cotton and the use of the backstrap loom are very ancient skills of the inhabitants of the northern coast of Peru, which continue to this day. These handicrafts are sold to tourists because there is a need to supplement family income [7]. However, handicrafts are part of

the production chain, which consists of producing native cotton to elaborate them in an artisanal way (by hand) and selling them to business people and tourists [8]. Handicrafts have an important role in the national economy, as they involve diverse indigenous peoples and important groups of women and are a highly inclusive activity.

Taking that into consideration, in order to start the reactivation of tourism, it was necessary for artisans to adhere to the preventive measures and comply with the established protocols [9]. Preventive measures against COVID-19, such as social distancing, self-isolation and travel restrictions, harmed the tourism sector, including the artisanal sector. The protocols point out measures aimed at maintaining social distancing, promoting hand hygiene, cleanliness and safety measures, and "monitoring" the health of participants in tourist events. The importance of health and the implementation of preventive measures can have a significant impact on the relational and spatiotemporal dimensions for the tourist, which are important factors to consider [10]. In a Nigerian study, participating artisans demonstrated their knowledge of social distancing, use of masks in public places, and compliance with personal hygiene measures to control COVID-19. However, they broke the strict shutdown rules imposed by the government due to their financial needs [9].

On the other hand, women artisans who work with native cotton living in the area surrounding the Historic Sanctuary Bosque de Pómac in the department of Lambayeque in Peru were severely affected by the preventive measures imposed by COVID-19, such as the lockdown. Native cotton was used by the pre-Columbian Moche, Sicán and Chimú cultures and has also been considered the ethnic-cultural genetic heritage of the nation and a flagship product of Lambayeque, as it is used as raw material for the production of fabrics [7]. However, cotton production currently needs to overcome challenges. There are not many sales; therefore, the profit does not compensate for the time invested [11], but they know this ancestral practice is an economic opportunity to maintain their culture.

At the beginning of 2022, the incipient post-COVID-19 tourist activity was significant, impacting artisans who worked with native cotton in Peru. Thanks to the decrease in lethality due to vaccination, governmental entities were promoting the reactivation of handicrafts throughout the country. On the other hand, the reactivation of the tourism economy due to COVID-19 is a means to promote sustainable growth and requires medium-term action. Latin American governments launched financial assistance programs with multilateral organizations [12]. There are three important areas to reactivate the economy: support for the incorporation of digital technologies, incentives for the formalization of companies, and biosafety protocols [13].

It should be noted that in rural areas, general prevention measures and the application of COVID-19 protocols for the reactivation of rural tourism and handicrafts were mostly not complied with. This was due to limited access to adequate training and interventions to reach this vulnerable group. Pregowska [14] indicates that distance learning can offer many advantages over traditional face-to-face teaching and training. Thus ref. [15], establishing actions for training in biosafety protocols for tourism ventures in the different territories of the country is of great importance. Virtual education allows participants to learn at any time of the day, anywhere, resulting in the spread of education to remote areas and societies with very little time for traditional education [14]. In the same way, for [16], the offer in higher education has been wasted in potentiating technological education that is also very useful for the tourism sector; with this, research gains strength because virtual educational intervention is a favorable practice to strengthen the capacities of students.

The present study aims to evaluate the level of compliance with preventive measures before and after performing a virtual educational intervention for craftswomen working with native cotton in the Lambayeque Region of Peru. As indicated by [14], distance education cannot replace practical workshops. However, it should be emphasized that, in situations where access to education is limited, distance learning provides an alternative, as has happened in the last two years due to the COVID-19 pandemic.

## 2. Theoretical Framework

### 2.1. Artisan Tourism

Today, the role of handicrafts globally has gained importance, both in developing countries and in rural areas in general. Local governments are making increasing efforts to promote rural development in order to prevent people from leaving rural areas. Handicrafts are considered an important tool for local economic development and job creation [17]. Clavellina-Miller [18] states that sectors such as trade and tourism have been the first to suffer the effects of the rapid expansion of the outbreak, whose effect on the global economy will be substantial.

The Sipán Artisanal Tourist Technological Innovation Center (Cite Sipán) in the Lambayeque Region indicated that the artisanal sector is represented by 94% women and 6% men, with the female quota the one that is mostly dedicated to this activity. For [19], the female gender plays an important role in the economic sustenance of the rural context. Women in these space areas are becoming true agents of change, as their ventures are increasingly sustainable, both in production processes and in the use of resources.

According to [20], social distancing, self-isolation, and travel restrictions have led to a reduction of the workforce in all economic sectors and the loss of many jobs, and the craft sector was no stranger to this situation. Fernandez et al. [17] argue that handicraft becomes a factor in alleviating poverty in the rural world. Therefore, the concept of handicrafts as a source of livelihood for poor rural regions, job creation, and the sustainability of the place is mainly emphasized. Ref. [21] indicates that artisans are then those people who use their specialized trade, especially those that involve working with hands, to discover or create, evaluate, and exploit opportunities to generate entrepreneurship.

Ratten et al. [21] indicate that interest in developing artisanal enterprises is growing throughout the global economy, thanks to the expansion of creative industries and the growing focus on homemade products and services with a cultural component, being important to preserve the cultural heritage of a region since authentic and original products made by artisans are part of the tourist experience. In this studio, the artisans work with native cotton (*Gossypium barbadense* L.), which is a fiber of natural colors whose use dates back about 5000 years and is characterized by being a plant of shrub development. In addition, it is not only the most resistant to dozens of pests and bacterial diseases, but it is resistant to high concentrations of soil salinity and drought and able to survive in sandy areas for up to five consecutive years without any irrigation [7].

Currently, the use of this fiber is used for the elaboration of traditional fabrics. They are characterized by being mainly based on local, regional, and tourist market demand. However, people who live from this type of business are very vulnerable to the economic downturn of the pandemic. The temporary closure of their activities causes a significant reduction in revenue [22].

The preventive measures against COVID-19 have allowed reactivation in many of the economic sectors, including tourism, and the artisanal sector is no stranger. Ecuador carried out an effective vaccination program that generated a significant decrease in infections and in demand for hospital beds [23]. Additionally, rural families in Ghana, as a preventive measure, used cloth masks, hand washing and increased the use of alcohol and bleach [24] so that the craft sector can be prepared for tourism. They must comply with preventive measures against COVID-19.

### 2.2. Virtual Education during COVID-19

Virtual education is a high-impact strategy for improving the coverage, relevance and quality of education at all levels and types of training due to its multimedia, hypertextual, and interactive characteristics [25].

Virtual education is an inclusive training process that seeks quality learning for all, which is based on recognizing that students learn differently [26]. It is important to consider accessibility in the virtual environment as it facilitates the learning process and interaction in the digitalization ecosystem of equal opportunities for everyone to participate [27].

Virtual education is an alternative form of learning that provides support to communities such as craftswomen who do not have easy access to training. In the context of COVID-19, the intervention carried out becomes a valuable strategy for them to prepare in order to reactivate tourism.

The pandemic affected the entire planet. Research has been developed in various disciplines as a result; however, this research does not identify its own theory, so these studies are empirical. Thus far, most of the attention in the literature has been paid to the health aspects of the pandemic [28].

In view of this situation, this research had the objective of evaluating the level of compliance with the general measures and the specific standards applied in the process of the artisan tourism activity before and after the virtual educational intervention for the reactivation of tourism in craftswomen working with native cotton. This also made it possible to determine the impact of the virtual intervention on the general prevention regulations of COVID-19 and the specific norms applied in the stages of the tourism activity process.

## 3. Materials and Methods

The research was a pilot quasi-experimental study with no control group. The population and sample was the totality of craftswomen who are dedicated to weaving with native cotton in the studio area, with 30 craftswomen from populated areas of Pomac III–Pitipo, Jotoro–Jayanca and La Raya–Túcume in the Lambayeque Region, which were selected for convenience at the Artisan Tourism Technological Innovation Center (Cite Sipán). Participants met the following inclusion criteria: they were of legal age and signed an informed consent form to participate in the study.

The observation technique was in person, visiting the craft workshops and house of the craftswomen, prior to telephone coordination. It was applied in order to collect data by means of a checklist on compliance with preventive measures against COVID-19 before and after the intervention. The checklist was prepared based on the instructions on sanitary measures before COVID-19 for the Peruvian artisanal sector [29]. Aiken's validity coefficient was used to test the validity of the instrument. For this purpose, there were 06 expert judges who belonged to the health sector and 4 to the tourism sector. Most of them are also professors, have a doctorate degree, and carry out research related to the subject under study. Each one made an assessment of 14 items qualifying yes = 1 point or no = 0 points, which were approved and statistically significant (Aiken's V = 1, $p = 0.001$).

The instrument contains 14 items with a two-dimensional yes or no response, detailing the general measures and specific regulations in the process of tourism activity before COVID-19, with the latter classified into three stages: before the handicraft production activity/process (stage 1), sale of handicrafts in the workshop, stand or store (stage 2), and after the handicraft activity has been carried out (stage 3).

The fieldwork to execute the study was carried out following the steps detailed below. Between October and December 2021, the checklist was applied to craftswomen working with native cotton. Then, the virtual educational intervention to prevent COVID-19 was applied, with the support of 5 podcasts and educational multimedia material on hand washing, social distancing, correct use of a mask, and digital means of payment. All this material was prepared for the intervention and stored at https://drive.google.com/drive/folders/16Ux7oLmS3NtedzetZn6609dDnDN7nhaB?usp=sharing.(accessed on 15 December 2022). The intervention was carried out after coordinating schedules with the craftswomen and lasted approximately three months, from March to May 2022. In addition, the handicraft workshops were implemented with signage. Subsequent to the intervention, the checklist was reapplied in June 2022. Likewise, in order to support the reactivation of tourism in this sector, two promotional videos were made that served to visualize the handicraft products made by the craftswomen and stored at: https://drive.google.com/drive/folders/1ymGBZoUfNY6wr-8Lke87pil6t67VtFjq?usp=sharing.(accessed on 15 December 2022) in order to help them market their handicrafts.

Tabulation and statistical analysis were performed with IBM SPSS version 25 using descriptive and inferential statistical techniques. Relative and absolute frequencies were calculated. The χ2 test was used in order to evaluate the variation before and after the intervention. The Pearson test was also used to identify the correlation since the data values obey a normal distribution according to the Kolmogorov normality test. This research has the approval of the Ethics and Research Committee of the Medicine School of the Universidad Católica Santo Toribio de Mogrovejo by means of Resolution No. 247-2021-USAT-FMED.

## 4. Results

The main findings of the fieldwork are described below in a differentiated way according to the blocks addressed.

Regarding the descriptive results of the sociodemographic profile of the craftswomen shown in Table 1, it is observed that the predominant age group is over 50 years old, which represents 50% of the population. The next most predominant age group is those from 41 to 50 years old (20%), and finally, the group of young craftswomen aged 20 to 30 years (13.3%). The marital status is married and cohabiting with 30% in each range. Only 26.6% have completed studies (elementary and high school level), and the number of family members is 2 to 3 in each range, representing 43.3%. Additionally, 90% of the craftswomen indicated that their income from the sale of handicrafts is less than the minimum living wage (SMV, by its Spanish acronym), which in Peru is PEN 1025.00.

**Table 1.** Sociodemographic profile of residents.

| Sociodemographic Variables | | No. | % |
|---|---|---|---|
| Age | 20–30 years old | 4 | 13.3 |
| | 31–40 years old | 5 | 16.7 |
| | 41–50 years old | 6 | 20.0 |
| | 51–51+ years old | 15 | 50.0 |
| Marital Status | Single | 8 | 26.7 |
| | Married | 9 | 30.0 |
| | Cohabitant | 9 | 30.0 |
| | Divorced | 3 | 10.0 |
| | Widowed | 1 | 3.3 |
| Education Level | Non-formal education | 2 | 6.7 |
| | Completed elementary school | 4 | 13.3 |
| | Uncompleted elementary school | 8 | 26.7 |
| | Completed high school | 8 | 26.7 |
| | Uncompleted high school | 4 | 13.3 |
| | Higher/technical education | 4 | 13.3 |
| Number of family members | 2–3 members | 13 | 43.3 |
| | 4–5 members | 11 | 36.7 |
| | 6–6+ members | 6 | 20.0 |
| Income from Crafts | <SMV | 27 | 90.0 |
| | =SMV | 2 | 6.7 |
| | >SMV | 1 | 3.3 |

*Own Elaboration*

The results show that 93.3% of the craftswomen have been vaccinated against COVID-19, which is the most effective prevention measure worldwide (Table 2).

**Table 2.** General preventive measures.

| Health Situation—COVID-19 | | No. | % |
|---|---|---|---|
| COVID-19 vaccine | Yes | 28 | 93.3 |
| | No | 2 | 6.7 |

Own elaboration.

The results show the level of compliance with the general measures before and after the virtual educational intervention by the craftswomen who work with native cotton weaving lines in the buffer zone of the Historic Sanctuary Bosque de Pómac (Table 3).

**Table 3.** Level of compliance with general measures before and after the intervention.

| Level of Compliance with General Preventive Measures | | No. | % |
|---|---|---|---|
| Before | Did not comply | 21.0 | 70.0 |
| | In process | 9.0 | 30.0 |
| After | In process | 10.0 | 33.3 |
| | Achieved | 20.0 | 66.7 |

Table 3 shows the evaluation of whether the craftswomen complied with the general preventive measures imposed by the Peruvian government: complete vaccination, hand washing, disinfection of areas, correct use of double surgical or a KN95 mask, and respecting a social distance of at least 1 m.

The results show that before the intervention, 70% of the craftswomen did not comply with the general preventive measures; however, after the intervention, 66% of them complied, and 33% of them were in the process of complying with the preventive measures before COVID-19.

In Table 4, the compliance with the specific standards applied in the stages of the tourism activity process before and after the intervention was analyzed, being as follows:

**Table 4.** Level of compliance with the specific standards applied in the stages of the tourism activity process before and after the intervention.

| Level of Compliance with the Specific Standards Applied in the Stages of the Tourism Activity Process | | | No. | % |
|---|---|---|---|---|
| BEFORE | Stage 1 | Beginning | 70.0 | |
| | | In Process | 13.3 | |
| | | Achieved | 16.7 | |
| | Stage 2 | Beginning | 100.0 | |
| | Stage 3 | Beginning | 76.7 | |
| | | In Process | 6.7 | |
| | | Achieved | 16.7 | |
| AFTER | Stage 1 | In process | 33.3 | |
| | | Achieved | 66.7 | |
| | Stage 2 | In process | 33.3 | |
| | | Achieved | 66.7 | |
| | Stage 3 | In process | 3.3 | |
| | | Achieved | 96.7 | |

Stage 1 analyzed whether the craftswomen cleaned and disinfected raw materials, inputs, tools and equipment at the beginning of the workday and whether they washed and disinfected their hands when they started producing their handicrafts. We also assessed whether, when more than two craftswomen meet to produce their handicrafts, they kept a social distance (1 m) and used a double surgical or a KN95 mask, with the following results before the intervention: 70% of craftswomen were present in the beginning level. The process level contained 13.3% of craftswomen, and only 16.7% of them had managed to comply with the preventive measures. After the educational intervention, the level of compliance with the preventive measures improved, and there were no craftswomen at the beginning level. Instead, there were now 33.3% at the process level, and 66.7% of the craftswomen managed to comply with the preventive measures for this stage.

Regarding compliance with stage 2, we examined whether the craftswomen installed and used a hand disinfection point at the entrance of the workshop, stand or craft store,

placed signage to promote social distancing at the point of sale, and used electronic means of payment or a digital wallet for transactions. We found that before the intervention, 100% of the artisans did not comply with the prevention measures and were at the beginning level; after the intervention, 33.3% of them were at the in-process level, and 66.7% had achieved these goals.

Stage 3 analyzed whether the craftswomen stored raw materials and inputs in the storage area, disposed of containers (bags, paper, etc.) and disinfected equipment and materials, and whether they washed and disinfected their hands at the end of the operation. The beginning level of compliance with these preventive measures contained 76.7% of the artisans. However, after carrying out the handicraft activity, 6.7% were at the process level, and 16.7% were at the achieved level. After the intervention, these results improved, and none of the artisans were at the beginning level; 3.3% were in process with this level, and 96.7% were at the achieved level.

In Figure 1, the correlational results show that before the intervention, the specific preventive measures in the three stages (before, during and after) of the development of the handicraft production and commercialization activity are closely related; in other words, each specific preventive measure carried out at each stage contributes to total compliance with the general rules, and the strength of this relation expressed by means of the chi-square test corroborates the a priori hypothesis that: "The greater the practice of the specific measures in stages 1, 2 and 3, the greater the compliance with the general standards as a whole."

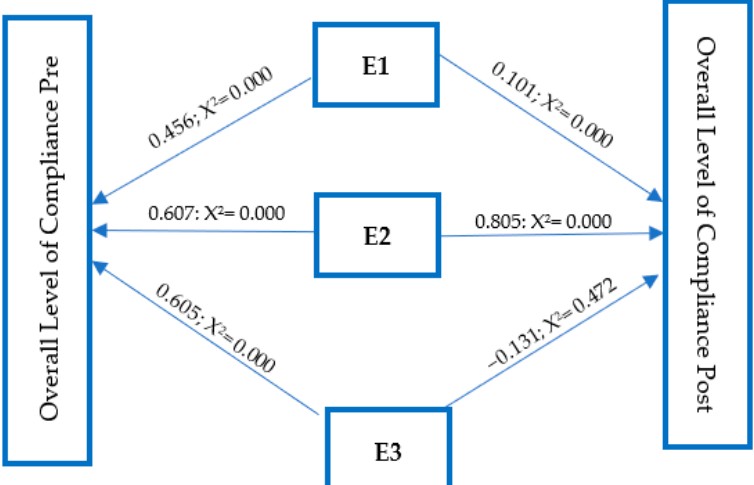

**Figure 1.** Determination of the relationship between the general preventive regulations of COVID-19 and the specific norms applied in the stages of the tourism activity process.

However, after applying the intervention and modifying behaviors to adapt to this new situation of reactivation, it is observed that the specific measures applied in two of the three stages demonstrate a greater benefit for increasing the general level of compliance with the general regulations. In other words, it is enough to apply the specific measures in the first (before) and second stage (during) to obtain a greater result in compliance with the general level of the regulations against COVID-19; therefore, the a priori hypothesis is modified as follows: "As long as the specific regulations are practiced in stages 1 and 2, compliance with the general rules of COVID-19 is ensured."

## 5. Discussion

As for the sociodemographic profile of the craftswomen of the native cotton weaving line who live in the towns of Pomac III–Pitipo, Jotoro–Jayanca and La Raya–Túcume in the Lambayeque Region, half of them are over 50 years old, while the least representative age group was between 20 and 30 years old. A little more than a quarter of them have



completed elementary and high school education. Currently, it can be observed that the children of the craftswomen are dedicated to other activities to improve their economic status. On the one hand, it is admirable, but analyzing it from the cultural legacy would jeopardize the transmission of this ancestral knowledge.

In addition, almost all of the craftswomen testify that their income from the sale of handicrafts is less than the minimum living wage (SMV), which in Peru is PEN 1025.00, equivalent to USD 262.82; in spite of the COVID-19 quarantine, this income was nil or very low for their family's subsistence. However, most craftswomen live with their spouses, whether married or cohabiting. In these areas, they are dedicated to family agriculture, and this economic activity gave them support during the pandemic. These families subsisted on what they produced in the fields. The closure of tourist activity affected the craftswomen because they stopped producing and selling their products; in other words, they could no longer earn income for their households. According to [30], the situation of most Pakistani craftswomen is similar, as they have been severely affected and face several problems due to the low sales of their products.

Additionally, almost all of the craftswomen have been fully vaccinated. Most craftswomen have been vaccinated because it is a requirement by the Peruvian government for the reactivation or reopening of tourism activities, and the study was developed in the context of the end of the third wave. Contrary to the study [31], reluctance to vaccinate is one of the main threats to the effectiveness of vaccination programs. The willingness to accept the COVID-19 vaccine among participants was lower in the third wave (34.8%) than in the first wave (44.2%). There were more concerns about vaccine safety in the third wave. Service or sales workers were less likely to accept the vaccine. This downward trend could also be the result of a high level of concern about vaccine safety. In the future, vaccination advertising should address these concerns, and an adequately and comprehensively tested vaccine would be helpful in gaining public confidence.

The results show that before the intervention, about two-thirds of the craftswomen did not comply with the general preventive measures before COVID-19; however, after the educational intervention with the support of podcasts and the implementation of signage in the places of elaboration and sales of handicrafts, two-thirds of the craftswomen complied with the preventive measures, and one-third were in the process of complying with the measures. In this regard [32], to reactivate tourism, they provided education and prepared promotional videos for the implementation of and compliance with health protocols before COVID-19, which were disseminated through the Zoom platform and a YouTube channel. They recommend that in order for tourists to comply with health protocols, they should be informed verbally and in writing through signs posted at the entrances about the mandatory requirement to wear a mask, wash their hands, and maintain a safe distance. In this way, they were able to strengthen compliance with health protocols to prevent the spread of COVID-19 in tourist sites.

In addition, a prerequisite for the craftswomen to reactivate their businesses was to have all the doses of the COVID-19 vaccine. According to [33], the attitude and efficacy of COVID-19 vaccination increased the level of hope and showed a strong ability to support tourism revival [34]. It is concluded that vaccination against COVID-19 increases the likelihood of travel; therefore, vaccination boosts tourism participation and, thus, the recovery of the tourism industry.

Meanwhile, ref. [35] showed that despite the delivery of vaccines, the applicability of COVID-19 pandemic preventive measures was too low, indicating that the rural area of Guraghe was at risk of contagion, which hinders the revival of tourism. However, ref. [36] indicated that vaccination coverage alone is not sufficient for the tourism industry to recover from the pandemic. Effective vaccination deployment is needed to substantially reduce the COVID-19 mortality rate.

Ref. [37] showed that educational intervention improves respondents' knowledge of the scientific basis and importance of preventive measures and their attitude towards vaccination. Therefore, an educational intervention on general COVID-19 prevention

measures was essential for native cotton craftswomen, not only regarding the call to comply with certain measures to prevent contagion but also to ensure adequate basic scientific literacy.

The protocols for reactivating tourism propose general measures such as ensuring social distancing, hand and respiratory hygiene procedures, and the disinfection of surfaces and areas. Likewise, actions were aimed at hygiene, the care and control of personnel (such as training or the organization of tasks and schedules), building conditions (cleanliness, arrangement of furniture, and circulation) and customer service (ways to facilitate access to information in pre- and post-sales, reception or accompaniment during activities, maximum and minimum participation quotas, among others) [10]. Some jobs in lodging, recreation, food services, and handicrafts should be incorporated into the new tourism normality, such as online payment and digital information, the development of protocols for visitors based on physiographic areas and visit dynamics, surveillance activities and the control of protection measures, hygiene management in common areas, and the collection and management of waste necessary to adapt the requirements of the service [38].

Those who are involved in the tourism sector have made an effort to be trained on the protocols for general measures before COVID-19. However, for small rural artisans, due to their lack of internet connection and budget, it was not as feasible to educate themselves in order to continue with their enterprise. Although general measures, vaccination, and the use of masks were reported through the media [39], the use of masks when leaving the house, covering the nose and mouth when sneezing or coughing, keeping a distance from others, washing hands with soap and water, and disinfecting object surfaces were the most frequently performed measures. One study revealed low adherence to preventive measures (physical distancing, use of a face mask, hand hygiene, covering the mouth when coughing/sneezing and avoiding touching the face) regarding COVID-19 [40]. In addition, social distancing rules are useful in preventing the spread of COVID-19, but the tourism industry is unlikely to revive with strict social distancing rules [41].

After applying the virtual educational intervention to the craftswomen, it was demonstrated that they improved their level of compliance with the specific measures in the first (before) and second stage (during) to obtain a greater result in compliance with the general level of the norms against COVID-19; therefore, the following a priori hypothesis arose: "As long as the specific standards are practiced in stages 1 and 2, compliance with the general COVID-19 standards is ensured."

The virtual educational intervention was necessary because these craftswomen needed to continue with their handicraft production, but they had to be careful because of the third wave of contagion, which was alarming. However, vaccination helped to reduce the lethality of the virus, which allowed the reactivation of tourism. The use of technology is a privileged tool in order to facilitate social distancing; for instance, it is suggested to obtain reservations, request payments or perform pre-check-ins or pre-check-outs through digital applications or websites [10]. In addition, digital marketing is a great tool to promote products and services on the network [41].

Female entrepreneurs face several challenges. It is known that, because of their poor origins, craftswomen are not familiar with digital media and cannot make their work known in places other than fairs and stores. This is a concerning fact since, due to the COVID-19 pandemic, fairs are no longer held, and stores cannot always remain open [42]. Despite the incorporation of new forms of payment, the difficulty with mobile phones and internet access makes this practice difficult to implement. On the other hand [43], maintaining operations during the global crisis due to COVID-19 was extremely difficult for small and medium-sized enterprises (SMEs), especially in the artisanal sectors. The most common problems were the limitations of direct marketing, which also involved paying bills, adopting an online platform, and setting up work from home.

Likewise, for [41], consumer behavior has shifted to more environmentally friendly products. Peruvian handicraft companies mainly offer ceramic products, handmade textiles, souvenirs made with gourds, religious sculptures made with chalk and plaster, objects

carved in wood, seeds, stones and palm bones, hats, objects made with leather, wax, metals, paintings by popular painters, and objects made with cow or bull horns. As a consequence, artisanal businesses can expand into e-commerce, innovation and ICT. This expansion makes these companies more competitive. Likewise, digital marketing is also a great tool to promote products and services on the network, reaching distant places. Technological advances, tools and the use of social networks are important in driving economic stability and growth. In financial crises and digital health, technological applications are at the forefront of addressing the undesirable impacts of COVID-19 on commercial activities, such as travel, leisure and tourism. This can be carried out, for example, with identity control and digital health passports, digital sales and payment, among others [44].

Finally, according to Ref. [45], there were several opportunities due to the pandemic to analyze and invest in critical organizational capacity-building and human capital through technological innovation, digital skills, and health protocols to generate resilience in the sector with a roadmap to respond to future shocks.

## 6. Conclusions

The craftswomen improved their compliance with general preventive measures and specific rules against COVID-19 after the virtual educational intervention. This has generated awareness among the craftswomen, favoring the reactivation of production and sales of their handicrafts. For this reason, it is recommended to continue with interventions to apply personal, family, and community care protocols to reactivate tourism activity.

The craftswomen adopted the prevention measures imposed by the government in the face of COVID-19 by complying with the quarantine and remaining in their homes, causing a decrease in their economic income; however, the Peruvian government provided several economic subsidies to help craftswomen and enterprises gradually reactivate their businesses, the sole purpose of which is to reactivate the economy through artisanal tourism in Peru.

Therefore, it is recommended that virtual training be maintained for the artisanal sector since it is an effective, economical and easily accessible method. For issues related to the improvement of their enterprises and health, the strengthening of capacities based on biosecurity protocols is crucial so that the visitor experience goes hand-in-hand with compliance with the regulations and elements of coexistence in the destination. Likewise, it is necessary that local governments not only implement virtual training workshops but also promote artisanal products through digital channels to generate visibility and reach alternative markets to facilitate economic reactivation.

## 7. Limitations

The present investigation was developed in the areas of La Raya–Túcume, Pómac III–Pitipo and Jotoro–Jayanca, with one of the limitations being the distance and accessibility of the workshops, stores, and houses of the artisans who work with native cotton. However, support was received from the Sipán Artisanal Tourist Technological Innovation Center–Cite Sipán for previous visits, information gathering, and applying the virtual educational intervention, and the response of the artisans was fruitful.

**Author Contributions:** Conceptualization and methodology, R.M.E.-H. and R.J.D.-M.; software, validation and investigation, R.M.E.-H., R.J.D.-M. and M.A.D.-V.; formal analysis, R.M.E.-H.; writing—original draft preparation, R.M.E.-H.; writing—review and editing, R.J.D.-M. and M.A.D.-V.; project administration, R.M.E.-H.; funding acquisition, R.J.D.-M. All authors have read and agreed to the published version of the manuscript.

**Funding:** This research received no external funding.

**Institutional Review Board Statement:** Not applicable.

**Informed Consent Statement:** Informed consent was obtained from all subjects involved in the study. The subjects were informed of their rights with the following legend: The fact that you answer this questionnaire implies your approval to participate in the present study. The data collected will be used only for academic purposes, and the confidentiality and anonymity of the data you provide will be respected at all times.

**Data Availability Statement:** The study has not reported any data.

**Acknowledgments:** The research study was developed with the support of the Universidad Catolica Santo Toribio de Mogrovejo within the framework of the Internal Teaching Research Contest 2021-II with Resolution -076-2021-USAT-RTDO.

**Conflicts of Interest:** The authors declare no conflict of interest.

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
