# Peer review of "Virtual Educational Intervention of Craftswomen Working with Native Peruvian Cotton during COVID-19 for Reactivating the Artisian Tourism"

_sustainability, doi:10.3390/su15075933_

Round 1

Reviewer 1 Report

Dear Authors,

Thank you for your paper and an enjoyable read. Here are some notes to assist in improving the paper:

Page 1, Lin14-15: "craftswomen has suffered the most" In what context, Peru? What about restaurants? Hotels? etc

Page 2, line 44: remove 'the'

Page 2, line: 76 'was remarkable' - what was?

Introduction: 'virtual intervention' is not clearly explained. Please review.

The methods section does elaborate more on virtual intervention protocols but it still needs to mentioned earlier.

The methods section is very clear and can be replicated easily.

The results section is clear although I could not test the data, I am assuming it is correct.

The conclusion is very short and has no limitations. Was the government correct in their covid-19 responses? Has the virtual intervention protocols now lapsed or can still be accessed? Perhaps this is a future study which haven't been mentioned either? 

I believe it is necessary to not just reset tourism and use intervention protocols to reset tourism. It needs to be acknowledged that the borders were closed to tourists therefore only local tourism could occur if any. This is either a limitation or needs to be weaved in the conversation of the paper as you need tourists to sell to. Also, most tourists buy these products when they see them and not necessarily in e-commerce stores. It should be worded that this is another way of sales but not their main way of selling traditional products.

Author Response

Respuesta al revisor

Dear review, thank you very much for allowing us revising and resubmitting tour article. We have found your comments to be highly helpful in improving the article in terms of the introduction, the theoretical background, the research methodology, and the theoretical contributions. After carefully reading your comments, we have introduced some changes in the manuscript to address your concerns. Following your recommendation, as shown in the new version of the manuscript. We present our detailed comments below.

RESPONSE TO REVIEWER

Comments:

Thank you for your paper and an enjoyable read. Here are some notes to assist in improving the paper:

Page 1, Lin14-15: "craftswomen has suffered the most" In what context, Peru? What about restaurants? Hotels? Etc

Response: Thank you very much for the comment, the improvements were made to the document.

Page 2, line 44: remove 'the'

Response: Thank you very much for the comment, the improvements were made to the document.

Page 2, line: 76 'was remarkable' - what was?

Response: Thank you very much for the comment, the improvements were made to the document.

Introduction: 'virtual intervention' is not clearly explained. Please review.

Response: In the document, a virtual educational intervention was added in the introduction and review of the literature to reinforce the topic.

The methods section does elaborate more on virtual intervention protocols but it still needs to mentioned earlier.

Response: Thank you very much for the comment, the improvements were made to the document.

The methods section is very clear and can be replicated easily.

Response: Thank you very much for the comment.

The results section is clear although I could not test the data, I am assuming it is correct.

Response: Thank you very much for the comment.

The conclusion is very short and has no limitations. Was the government correct in their covid-19 responses? Has the virtual intervention protocols now lapsed or can still be accessed? Perhaps this is a future study which haven't been mentioned either? 

Response: A paragraph was added in the conclusions responding to the observation made.

I believe it is necessary to not just reset tourism and use intervention protocols to reset tourism. It needs to
be acknowledged that the borders were closed to tourists therefore only local tourism could occur if any.
This is either a limitation or needs to be weaved in the conversation of the paper as you need tourists to
sell to. Also, most tourists buy these products when they see them and not necessarily in e-commerce
stores. It should be worded that this is another way of sales but not their main way of selling traditional
products.

Response: Thank you very much for the comment.

Reviewer 2 Report

Well written paper focused on interesting topic  - suggestions are following: 

Literature review should include more theoretical sources related with the craftwork, craftworkers and women aspect / reputable and newer date literature units 

Better presentation of research questions or hypothesis, results of testing

Better explanation of Figure 1

Explanation: limitation of the study and future research?

Author Response

Response to Reviewer

Dear review, thank you very much for allowing us revising and resubmitting tour article. We have found your comments to be highly helpful in improving the article in terms of the introduction, the theoretical background, the research methodology, and the theoretical contributions. After carefully reading your comments, we have introduced some changes in the manuscript to address your concerns. Following your recommendation, as shown in the new version of the manuscript. We present our detailed comments below.

RESPONSE TO REVIEWER

Comments:

Well written paper focused on interesting topic - suggestions are following:

Literature review should include more theoretical sources related with the craftwork, craftworkers and women aspect / reputable and newer date literature units.

Response: Thank you very much for the comment. In the document was added in the introduction and review of the literature to reinforce the topic.

Better presentation of research questions or hypothesis, results of testing.

Response: Thank you very much for the comment. In the document was added in the introduction and review of the literature to reinforce the topic.

Better explanation of Figure 1

Response: Thank you very much for the comment, the improvements were made to the document.

However, after applying the intervention, modifying behaviors to adapt to this new situation of reactivation, it is observed that the specific measures applied in two of the three stages are with greater benefit to increase the general level of compliance with the general regulations, in other words, it is enough to apply the specific measures in the first (before) and second stage (during),  to obtain a greater result in compliance with the general level of the regulations against COVID 19, therefore the a priori hypothesis is modified: "As long as the specific regulations are practiced in stages 1 and 2, compliance with the general rules of COVID 19 is ensured."          

Explanation: limitation of the study and future research?

Response: Thank you very much for the comment, the improvements were made to the document.

Reviewer 3 Report

As per attachment

Author Response

Dear review, thank you very much for allowing us revising and resubmitting tour article. We have found your comments to be highly helpful in improving the article in terms of the introduction, the theoretical background, the research methodology, and the theoretical contributions. After carefully reading your comments, we have introduced some changes in the manuscript to address your concerns. Following your recommendation, as shown in the new version of the manuscript. We present our detailed comments below.

Round 2

Reviewer 1 Report

Dear Authors,

You should be commended on this revision. All concerns appear to have been addressed and the focus of the paper is now more accurate in its title, literature review, discussion and conclusion. Please proof-read one more time.

Author Response

Dear reviewer: We thank you for your time and comments.

Reviewer 3 Report

All corrections have been made based on the comments and satisfactory 

Author Response

(The authors gave the same response as above.)
